# Therapeutic Plasma Exchange in Corticosteroid-Refractory Multiple Sclerosis Relapses: Mechanisms, Efficacy, and Integration into Clinical Practice

**DOI:** 10.3390/biomedicines13102399

**Published:** 2025-09-30

**Authors:** Mariano Marrodan, Maria C. Ysrraelit, Jorge Correale

**Affiliations:** 1Departamento de Neurología, Fleni, Buenos Aires C1428AQK, Argentina; mmarrodan@fleni.org.ar (M.M.); mcysrraelit@fleni.org.ar (M.C.Y.); 2Departamento de Química Biológica, Instituto de Química y Fisicoquímica Biológicas (IQUIFIB), Facultad de Farmacia y Bioquímica, Universidad de Buenos Aires-CONICET, Buenos Aires C1121ABG, Argentina

**Keywords:** multiple sclerosis, relapses, corticosteroid-refractory MS, plasmapheresis, therapeutic plasma exchange, humoral immune responses

## Abstract

Therapeutic plasma exchange (TPE) is increasingly recognized as a critical escalation therapy for managing acute multiple sclerosis (MS) relapses refractory to high-dose corticosteroids. Neuropathological and clinical evidence implicate humoral immune mechanisms, particularly autoantibodies, immune complexes, and complement activation, as key pathogenic drivers in a subset of MS attacks, notably those consistent with immunopathological pattern II. By removing these circulating immune effectors, TPE provides a rational strategy to dampen inflammation and promote neurological recovery. This review integrates current mechanistic insights with clinical efficacy data and practical implementation strategies for TPE in corticosteroid-refractory MS. Evidence from randomized controlled trials and observational cohorts demonstrates moderate-to-marked functional improvement in 40–60% of patients, with the greatest benefit observed when therapy is initiated within 14 days of symptom onset and cases demonstrating active inflammatory lesions on MRI. Predictors of a favorable response include younger age, short disease duration, severe syndromes involving optic nerve, brainstem, or spinal cord, and CSF markers of intrathecal B-cell activity. Although TPE is generally well tolerated in experienced centers, its broader adoption of TPE is limited by variability in access, institutional protocols, and provider familiarity. Standardization of treatment algorithms, validation of predictive biomarkers, and incorporation into streamlined clinical pathways are critical to maximizing its clinical impact. Future priorities include comparative trials against alternative escalation therapies, biomarker-guided patients’ selection, and comprehensive health-economic evaluations. Taken together, current evidence and recommendations from major neurology and apheresis societies support TPE as a valuable therapeutic modality capable of significantly improving relapse outcomes in appropriately selected MS patients.

## 1. Introduction

Management of acute relapses in multiple sclerosis (MS) remains a cornerstone of disease intervention. Acute exacerbations can result in substantial and sometimes irreversible disability, if inadequately managed [1,2]. High-dose intravenous corticosteroids most commonly methylprednisolone at 1 g/day for 3–5 days, remain the first-line treatment [3]. Their efficacy derives from multiple mechanisms, including inhibition of leukocyte migration, modulation of cytokine expression, attenuation of blood–brain barrier permeability, and induction of apoptosis of activated T cells within inflammatory plaques [4].

However, 20% to 40%, either fail to achieve adequate clinical improvement or develop adverse effects that preclude full treatment. Responsiveness may be influenced by such a lesion burden, underlying immunopathological subtype, and treatment delay. In this context alternative interventions capable of attenuating inflammation and promoting recovery are essential [5]. Importantly, plasmapheresis/therapeutic plasma exchange (TPE) is not considered a standard first-line therapy, even in severe relapses, but is endorsed by major guidelines as a second-line escalation strategy specifically in cases refractory to high-dose corticosteroids.

Therapeutic plasma exchange, also known as plasmapheresis, has emerged as a viable second-line treatment option. TPE operates through the extracorporeal removal of circulating pathogenic autoantibodies, immune complexes, complement components, and inflammatory mediators [6,7,8,9]. These elements are central to the humoral immune pathology underlying a subset of MS relapses, particularly those consistent with [10] antibody-mediated pathology [7,11]. As such, TPE represents a rational, targeted intervention for patients who do not respond to corticosteroids.

This narrative review synthesizes current evidence on the mechanistic basis, clinical efficacy, safety profile, and practical implementation of TPE in the treatment of corticosteroid-refractory MS relapses. It further identifies key knowledge gaps and delineates future research priorities required to optimize its integration into clinical practice.

## 2. Mechanistic Rationale for TPE in MS

The immunopathology of MS relapses is heterogeneous, reflecting contributions from both cellular and humoral immune components. Traditionally, attention has focused on T-cell-mediated inflammation, particularly involving autoreactive CD4+ T cells, of the Th1 and Th17 lineages, which infiltrate the central nervous system (CNS) and release pro-inflammatory cytokines such as IFN-γ, IL-17, and TNF-α. These cytokines disrupt the blood–brain barrier (BBB), recruit additional immune cells, and activate resident microglia [12]. Activated microglia further amplify tissue injury through the release of cytotoxic mediators, including nitric oxide, reactive oxygen species (ROS), and matrix metalloproteinases, thereby amplifying axonal damage and demyelination [13,14]. Over the past two decades, however, accumulating evidence has highlighted the pivotal role of B cells and humoral immune mechanisms in MS pathogenesis [15]. Neuropathological studies have identified a subset of lesions, so-called “pattern II” lesions, characterized by substantial immunoglobulin and complement deposition, implicating antibody- and complement-mediated cytotoxicity as key drivers of demyelination in these cases [16].

As complement passively diffuses from the bloodstream, and circulating antibodies arise from the periphery thus, TPE is uniquely suited positioned to target this pathogenic pathway by removing these circulating immune effectors.

B cells are also detectable in the cerebrospinal fluid (CSF) and enriched within meningeal and perivascular infiltrates in the CNS [17,18]. Beyond their classical role as antibody-producing cells, B lymphocytes contribute to disease pathogenesis through antigen presentation and cytokine secretion that modulate T-cell responses [19]. In a subset of patients, clonally expanded B cells persist intrathecally, and their antibody products have been associated with direct cytotoxic effects on oligodendrocytes [20,21]. While TPE does not directly reduce intrathecal Ig synthesis, modulation of peripheral immune activity may secondarily influence compartmentalized CNS inflammation. This indirect effect, although less well established, provides a potential link between peripheral immune clearance and central immune modulation [7,22]. In addition to removing humoral mediators, TPE may exert broader immunomodulatory effects. It has been shown to alter peripheral cytokine profiles, reduce the activation of circulating leukocytes, and downregulate adhesion molecules involved in CNS trafficking [23]. Recent immunophenotyping studies demonstrate that TPE decreases pathogenic Th1 cells and CD11c+ B cells, both implicated in MS pathogenesis and correlate with clinical response to TPE [24]. Moreover, there is emerging interest in how peripheral immune modulation through TPE might influence central immune mechanisms. Although direct evidence is limited, it is hypothesized that removing peripheral immune stimuli can downregulate CNS microglial activation, reduce antigen presentation, and dampen the neuroinflammatory milieu [25]. By resetting the peripheral immune environment, TPE may create conditions favorable for endogenous repair processes and neuroprotection [26,27].

Collectively, these mechanistic insights provide a robust biological foundation for the use of TPE in MS relapses characterized by humoral immune predominance. Stratification of patients based on clinical and radiological features may enable more precise therapeutic targeting, maximizing the likelihood of significant benefit.

## 3. Methodological Approach

This narrative review critically evaluates the role of TPE in the management of MS relapses, with particular focus on cases refractory to high-dose corticosteroid therapy. A comprehensive search strategy was implemented across PubMed, Scopus, and Web of Science databases, employing a combination of Medical Subject Headings (MeSH) and free-text terms: “multiple sclerosis,” “relapse,” “plasmapheresis,” “therapeutic plasma exchange,” “corticosteroid-refractory,” and related synonyms. The search was limited to studies published in English and involved adult human subjects.

Eligibility criteria included studies enrolling adult patients diagnosed with either relapsing remitting MS (RRMS) or progressive MS experiencing acute relapses in whom TPE was administered following inadequate clinical response to standard high-dose intravenous methylprednisolone (IVMP). To ensure diagnostic accuracy, episodes were required to represent true MS relapses, with exclusion of pseudo-exacerbations (e.g., infection, fever, metabolic imbalance) and carefully considering alternative diagnoses such as neuromyelitis optica spectrum disorder (NMOSD) or myelin oligodendrocyte glycoprotein antibody-associated disease (MOGAD). Only studies employing validated clinical outcome measures, such as the Expanded Disability Status Scale (EDSS), visual acuity assessments, or detailed functional system scoring, were included. A minimum follow-up of four weeks post-TPE was required to evaluate sustained clinical response. Studies focusing exclusively on NMOSD, MOGAD, or pediatric populations were excluded, unless they provided a distinct subgroup analysis for MS patients.

In addition to peer-reviewed data, we examined recent consensus statements and treatment algorithms from major professional societies, including the American Academy of Neurology (AAN) [28], and the American Society for Apheresis (ASFA) [29], to contextualize findings within current clinical practice recommendations.

## 4. Clinical Efficacy of Therapeutic Plasma Exchange

Therapeutic plasma exchange has emerged as a critical escalation therapy for acute MS relapses refractory to high-dose IVMP. A growing body of clinical evidence, including randomized controlled trials, large-scale observational cohorts, and meta-analyses, supports its efficacy in promoting functional recovery, particularly in relapses driven by humoral immune mechanisms (Table 1).

The pivotal evidence for TPE’s efficacy in corticosteroid-refractory demyelinating disease derives from the seminal randomized, sham-controlled trial conducted by Weinshenker et al. [30]. This study enrolled 22 patients with acute CNS demyelinating events, including MS and related disorders, who had failed to a 5-day course of IVMP. Participants received either active TPE (seven sessions over 14 days) or sham apheresis. Moderate to marked neurological improvement was observed in 42% of patients treated with TPE, compared with only 6% in the control group. This landmark trial provided Class I evidence of efficacy and underscored the pathogenic relevance of antibody-mediated mechanisms in a subset of steroid-refractory demyelinating syndromes. It is important to interpret these findings in historical context. At the time of this trial, diagnostic biomarkers such as AQP4-IgG and MOG-IgG were not yet available, and patient cohorts often encompassed heterogeneous demyelinating disorders, including neuromyelitis optica spectrum disorder (NMOSD), acute disseminated encephalomyelitis (ADEM), and Marburg variants, in addition to MS. As a result, contamination by non-MS cases is highly probable, and conclusions regarding efficacy in definite MS should be considered with caution.

Subsequent retrospective and prospective cohort studies have corroborated these findings across broader clinical contexts (Table 1). RRMS attacks were rare in these cohorts, dominated by severe cases such as NMOSD, MOGAD, or ADEM. This imbalance likely facilitated the demonstration of therapeutic benefit in those disorders compared with typical MS relapses. Indeed, attempts to reproduce the efficacy of TPE in strictly MS cohorts, such as the study by Brochet et al., largely failed to confirm significant benefit, an outcome that may reflect both the less severe nature of many RRMS relapses and the methodological challenges of assembling adequately powered, homogeneous MS-only cohorts [31].

Nonetheless, several well-designed analyses have provided meaningful support for TPE in MS. In a retrospective series of 59 patients with steroid-unresponsive demyelinating attacks, most of whom had MS, Keegan et al. [32] reported moderate to marked neurological improvement in 44%. of patients. Similarly, larger cohorts, including a German registry-based study of more than 300 patients, demonstrated comparable response rates (~45%), with particularly favorable outcomes in cases of acute optic neuritis, transverse myelitis, and brainstem syndromes. Collectively, these studies identify rapid and severe relapses, shorter disease duration, and early initiation of TPE as predictors of a favorable response [33].

The timing of TPE initiation is a critical determinant of therapeutic efficacy. Most studies indicate that administration within 7 to 14 days of relapse onset is associated with significantly improved outcomes, whereas delays beyond 21 to 30 days correlate with diminished efficacy, likely reflecting irreversible tissue damage and waning inflammatory activity [34].

Clinical outcomes are most commonly evaluated using the Expanded Disability Status Scale (EDSS), visual acuity measurements, and motor strength testing, depending on the affected domain. Data suggest that partial to complete recovery can be achieved in up to 60–70% of cases when TPE is initiated early in the course of a steroid-refractory relapse. For example, one cohort reported 60.9% good or full recovery, while overall response rates in relapsing–remitting MS or clinically isolated syndromes (CIS) range from ~59–87%. Compared with steroids alone, TPE confers a significantly higher likelihood of achieving marked functional improvement, with recovery typically stabilizing within 4–8 weeks, although continued gains may occur beyond this period [35,36,37,38].

These findings underscore the substantial and durable therapeutic benefit of TPE in carefully selected patients experiencing corticosteroid-refractory MS relapses. Nevertheless, considerable heterogeneity in study designs, treatment protocols, patient inclusion criteria, timing of intervention, and outcome assessments limits the ability to perform direct cross-study comparisons. This highlights the urgent need for standardized response criteria and prospective, well-controlled trials to validate efficacy across diverse clinical contexts.

In conclusion, TPE should be regarded as an effective rescue therapy for acute MS relapses unresponsive to corticosteroids, particularly when initiated early and in patients with radiological or immunological features suggestive of antibody-mediated mechanisms. Its favorable benefit-risk profile supports its adoption as a standard second-line intervention in the management of severe, steroid-refractory MS exacerbations. It must be emphasized, however, that TPE is not a standard first-line treatment for MS relapses, even when severe; rather, it is endorsed by current guidelines as a second-line escalation strategy in cases that are refractory to high-dose corticosteroids. Importantly, these data apply to acute inflammatory attacks and should not be extrapolated to chronic progression in primary or secondary progressive MS. At present, there is no high-level evidence supporting TPE for non-relapsing progression, and its use in progressive phenotypes should be reserved for superimposed, clearly inflammatory relapses (e.g., new gadolinium-enhancing lesions) while recognizing this as a priority area for future prospective study.

**Table 1 biomedicines-13-02399-t001:** Summary of key clinical studies evaluating therapeutic plasma exchange in MS relapses refractory to corticosteroids ^†^.

Study	Design	N	Population	Timing to TPE, Median in Days *	Response Rate (%)	Outcome Measure
Weiner et al., 1989 [39]	RCT/sham controlled as adjunctive therapy	76	RRMS	≤2	64	DSS
Weinshenker et al., 1999 [30]	RCT/sham- controlled crossover	22	Mixed demyelinating diseases	≤14	42	TND
Keegan et al., 2002 [32]	Retrospective	59	Mixed demyelinating diseases	17	44	TND
Llufriu et al., 2009 [34]	Retrospective	18	RRMS	27	28 **	EDSS
Ehler et al., 2015 [35]	Retrospective	37	RMS	44	81.1	EDSS
Correia et al., 2018 [36]	Retrospective	46	RMS	33 ***	80.4	EDSS
Marrodan et al., 2021 [37]	Retrospective	23	RRMS	15 ***	78	EDSS
Blechinger et al., 2021 [9]	retrospective	118	RMS	39	78.8	EDSS
Bunganic et al., 2022 [8]	Retrospective	155	RRMS	49	50	EDSS
Iacono et al., 2024 [7]	Retrospective	59	Mixed neuroimmunological diseases	26	80 ****	EDSS and MRS
Mesaros et al., 2024 [38]	Retrospective	107	RMS	32	80.9	EDSS

^†^ Some historical datasets include pooled acute demyelinating syndromes (MS together with NMOSD/ADEM); where available, MS-only series (e.g., Ehler 2015 [35]; Correia 2018 [36]; Marrodan 2021 [37]; Blechinger 2021 [9]) are highlighted in the table and cited in the text. DSS: Kurtzke Disability Status Scale; EDSS: Expanded Disability Status Scale; MRS: Modified Ranking Scale; RCT = Randomized clinical trial; RMS: Remitting Multiple Sclerosis; RRMS: Relapsing-remitting Multiple Sclerosis; TND = targeted neurological deficits: coma, aphasia, acute severe cognitive dysfunction, hemiplegia, paraplegia, or quadriplegia. TPE: Therapeutic Plasma Exchange. * When available ** 28% improvement at discharge 55% improvement at 6 months. *** Mean. **** in MS patients.

## 5. Predictors of Therapeutic Response

Identifying the clinical, radiological, and biological predictors of therapeutic response to TPE is essential for optimizing patient selection and treatment timing in corticosteroid-refractory MS relapses. Although TPE is an effective second-line intervention, outcomes vary considerably among individuals. A growing body of evidence has highlighted several key factors associated with favorable and unfavorable responses (Table 2). Refining these parameters, particularly when integrated with emerging biomarker data, may enable a more personalized application of TPE and improve clinical outcomes in this high-risk MS population.

### 5.1. Age and Disease Duration

Younger age and shorter disease duration are consistently associated with a higher likelihood of benefit from TPE. This trend likely reflects a more active and predominantly inflammatory disease phenotype in earlier stages, with greater preservation of axons and oligodendrocytes. In contrast, patients with longstanding disease are more prone to chronic neurodegenerative changes, such as axonal transection, gliosis, and remyelination failure, that are less responsive to immunomodulatory or immunodepleting interventions [9]. Retrospective studies have demonstrated that age below 40–45 years and disease duration under 5 years are independent predictors of moderate-to-marked neurological improvement following TPE [40,41]. These findings suggest that younger patients not only retain greater neuroplastic potential but also have relapses more likely driven by active humoral mechanisms, thereby increasing responsiveness to TPE.

### 5.2. Relapse Phenotype and Severity

The anatomical localization and severity of the relapse are strong determinants of responsiveness to TPE. Severe relapses affecting functionally eloquent regions, particularly optic neuritis, transverse myelitis, and brainstem syndromes, are associated with a greater likelihood of functional recovery following treatment. In optic neuritis, early TPE has been associated with substantial improvement of visual acuity, particularly in patients with profound initial deficits. However, these findings are largely derived from pooled cohorts including MS and NMOSD, warranting caution in extrapolation to MS alone [42,43]. Similarly, in partial or complete transverse myelitis, TPE has been linked to improvements in motor and sphincter function, even in patients who failed to respond to high-dose corticosteroids [28,44].

### 5.3. MRI Features of Active Inflammation

Neuroimaging markers, particularly gadolinium-enhancing lesions on brain or spinal cord MRI, are among the most reliable predictors of TPE efficacy. These lesions signify ongoing inflammation, BBB disruption, and active leukocyte trafficking [36].

In a multicenter study the presence of contrast enhancement was associated with significantly higher odds of clinical improvement following TPE [33]. By contrast, patients with non-enhancing, T2-hyperintense lesions, more indicative of chronic, irreversible pathology, were far less likely to derive benefit [40]. These observations reinforce the concept that TPE efficacy is closely linked to reversibility of tissue injury and the temporal dynamics of immune-mediated demyelination.

### 5.4. Timing of Intervention

Time from relapse onset to TPE initiation represents the most modifiable predictor of response. The therapeutic window for optimal benefit appears to be within the first 7–14 days, with efficacy declining sharply beyond 21 days. This time sensitivity underscores the importance of early recognition of steroid-refractoriness and prompt referral for apheresis [9,26].

Delayed initiation may permit secondary axonal degeneration and irreversible structural damage, thereby limiting the potential for recovery even if the immunopathological drivers are effectively removed [33].

### 5.5. Baseline Neurological Status and Chronic Deficits

Several baseline factors are consistently associated with limited or absent response to TPE. These include:-Poor baseline neurological function (e.g., EDSS ≥ 7.5) at relapse onset, is strongly predictive of poor outcomes and likely reflects extensive, irreversible tissue [34].-Absence of MRI inflammatory activity, such as lack of gadolinium enhancement or diffusion restriction, has been linked to minimal clinical benefit [35].-Presence of fixed neurological deficits from prior relapses can further complicate assessments of new inflammatory activity and limit recovery potential.-Slowly progressive symptoms evolution rather than abrupt relapses onset, often indicates a non-inflammatory or degenerative process (e.g., progressive MS) that is inherently less responsive to immunomodulatory interventions such as TPE [45].

### 5.6. CSF and Serological Biomarkers

The identification of predictive biomarkers offers considerable promise for refining patient selection and optimizing the efficacy of TPE in corticosteroid-refractory MS relapses. Patients exhibiting a predominant humoral immune signature are thought to derive the greatest benefit from this intervention. While histopathological classification of lesion subtypes is not feasible in routine clinical practice, CSF biomarkers reflecting B-cell activation provide practical surrogates for humoral-driven pathology. Relevant markers include the presence of oligoclonal IgG bands, elevated intrathecal synthesis of immunoglobulin, particularly IgM isotypes, increased concentrations of free κ and λ light chains, and upregulation of B cell–activating pathways such as the BAFF/APRIL axis or the chemokine CXCL13 [46]. Together, these indicators reflect intrathecal B-cell maturation and heightened humoral immune activity. Integration of such biomarkers into clinical decision-making may help identify patients most likely to respond to TPE, thereby enabling a more judicious and personalized application of this second-line therapy.

## 6. Safety Profile of TPE

Therapeutic plasma exchange is widely regarded as a safe and generally well-tolerated intervention when performed by adequately trained personnel in specialized centers. Its safety profile has been well documented across neurological, nephrological, and hematological disorders, with a low incidence of serious adverse events in appropriately selected patients [47]. Nonetheless, TPE is an invasive procedure that induces significant physiological shifts, including fluid and electrolyte imbalances, immunoglobulin depletion, and transient hemodynamic alterations. Accordingly, careful patient selection, comprehensive pre-procedural evaluation, and vigilant intra- and post-procedural monitoring are essential to minimize risk.

In the MS setting, where TPE is typically employed as a second-line intervention for severe or corticosteroid-refractory relapses, therapeutic benefit must be judiciously weighed against procedural risks, particularly in individuals with advanced disability or relevant comorbidities. Table 3 summarizes the principal adverse events associated with TPE and outlines evidence-based strategies for their prevention and management, thereby enhancing procedural safety and optimizing patient outcomes.

### 6.1. Common and Anticipated Adverse Effects

Most adverse effects associated with TPE are mild, transient, and arise from the extracorporeal circulation or the anticoagulant used during the procedure:-Hemodynamic instability: Hypotension is among the most frequently reported side effects, often resulting from rapid intravascular volume shifts, autonomic dysfunction in neurologically impaired patients, or an inadequate compensatory response to volume replacement. Risk can be mitigated by employing slower exchange rates, ensuring adequate pre-procedural hydration, and judicious use of vasopressors when necessary [7].-Electrolyte imbalances: Hypocalcemia related to citrate anticoagulation is common, presenting with perioral paresthesias, muscle cramps, or in severe cases, arrhythmias. Prophylactic calcium supplementation (e.g., calcium gluconate infusion) during the procedure and close monitoring of ionized calcium levels are effective preventing [48,49].-Gastrointestinal and constitutional symptoms: Nausea, vomiting, dizziness, chills, and fatigue may occur particularly during initial sessions. These events are generally self-limiting and respond well to symptomatic management [49].

### 6.2. Vascular Access and Catheter-Related Risks

Vascular access is a critical determinant of procedural safety in TPE. In most neurological patients, peripheral venous access is preferred; however, the use of central venous catheters (CVCs) is often required to achieve flow rates compatible with therapeutic efficacy:-Catheter-associated thrombosis and infection: non-tunneled CVCs carry an increased e risk of catheter-associated bloodstream infections, particularly in immunosuppressed individuals. Preventive strategies include strict aseptic techniques, use of antimicrobial-impregnated catheters, and minimizing catheter dwell time [50,51].-Mechanical complications: Pneumothorax, hemothorax, or arterial puncture are rare but serious complications during CVC insertion [52]. The adoption of ultrasound guidance has significantly reduced these risks and is now considered standard of care [53].-Hemorrhagic events: Although uncommon, bleeding complications may occur secondary to heparin anticoagulation or depletion of coagulation factors [49]. Regular monitoring of coagulation parameters and avoidance of concurrent anticoagulation, unless clearly indicated, are recommended.

### 6.3. Immunologic and Allergic Reactions

TPE may elicit immunologic responses, particularly related to the type of replacement fluid used:-Albumin vs. Fresh Frozen Plasma (FFP): Albumin is generally preferred in MS due to a lower risk of hypersensitivity reactions. FFP is reserved for specific indications (e.g., thrombotic thrombocytopenic purpura) but carries a higher risk of anaphylaxis and transfusion-related lung injury [54,55].-Allergic manifestations: Mild reactions including urticaria, pruritus, or flushing are relatively common and usually manageable with antihistamines. Severe reactions such as anaphylaxis are exceedingly rare but require immediate discontinuation of the procedure and emergency management [49].

### 6.4. Serious and Rare Adverse Events

Although uncommon, serious complications have been reported and warrant high clinical vigilance: [56,57,58].

-Sepsis and bloodstream infections: Particularly in elderly or immunocompromised individuals, bacteremia may result from catheter colonization or manipulation. Early recognition and empiric antimicrobial therapy are critical.-Coagulopathies and bleeding: Repeated exchanges may deplete clotting factors, underscoring the need for periodic fibrinogen monitoring and, in select cases, the administration of FFP or cryoprecipitate.-Metabolic derangements: Electrolyte abnormalities such as hypokalemia, hypomagnesemia, or hypernatremia may occur, especially in patients with underlying renal dysfunction or when large fluid volumes are administered. Careful monitoring and targeted replacement are required.

### 6.5. Long-Term Safety and Immunologic Tolerance

Cumulative evidence from MS cohorts indicates that repeated or periodic TPE does not confer increased risk of sustained immunosuppression, opportunistic infections, or secondary autoimmune disorders. Although TPE removes both pathogenic and protective humoral components, the immune system rapidly reconstitutes immunoglobulins and complement factors, typically within days to weeks.

Overall, while TPE carries a defined spectrum of potential adverse effects, the majority are predictable, manageable, and infrequent when procedures are performed in experienced, high-volume centers following established protocols. Given its favorable long-term safety profile, TPE remains a viable and often underutilized option for the management of severe or corticosteroid-refractory MS relapses [34,59].

## 7. Practical Considerations and Protocol Variability

Therapeutic plasma exchange protocols for the management of corticosteroid-refractory MS relapses are primarily informed by expert consensus and clinical experience, rather than uniformly validated by large-scale randomized trials. Standard regimens typically involve 5 to 7 apheresis sessions performed on alternate days, with each session exchanging approximately 1 to 1.5 plasma volumes [29]. This schedule is designed to maximize the clearance of circulating pathogenic immunoglobulins, immune complexes, complement components, and other soluble pro-inflammatory mediators implicated in MS relapses pathogenesis. The full course is completed within 10 to 14 days, although adaptations are often made based on individual patient response, adverse event profiles, and institutional logistics [32].

Albumin (5%) is the replacement fluid of choice due to its excellent safety profile and low incidence of hypersensitivity reactions. In select clinical scenarios, such as coagulopathies, hepatic insufficiency, or significant bleeding risk, FFP may be used either partially or fully to restore clotting factors. However, FFP administration requires vigilant monitoring for transfusion-related complications, including allergic reactions, volume overload, and electrolyte disturbances, particularly in patients with underlying cardiopulmonary or renal comorbidities [60].

### Strategies to Optimize Implementation

Efforts to standardize and expand TPE availability in MS have increasingly focused on harmonization of protocols, clarification of treatment criteria, and innovative models of care delivery. Key strategies include:-Protocol Harmonization: Development and dissemination of standardized TPE protocols through professional societies such as the ASFA and the AAN, which support consistent practice and facilitate multicenter research [28,29].-Defined Criteria for TPE Initiation: Incorporation of explicit definitions of corticosteroid non-response and clear timing thresholds for TPE initiation into relapse management algorithms ensuring timely escalation during the critical therapeutic window [8].-Streamlined Referral Pathways: Implementation of fast-track care models that enable rapid transitions from outpatient neurology services to apheresis centers, minimizing treatment delays and optimizing recovery potential [61].-Innovative Access Models: Deployment of telemedicine-based consults and mobile apheresis units to extend access in resource-limited regions, thereby reducing disparities in access to advanced immunomodulatory care [62].

## 8. Gaps in Current Knowledge

Despite accumulating evidence supporting the role of TPE in corticosteroid-refractory MS relapses, important gaps persist that limit both its clinical application and its integration into treatment guidelines. These gaps highlight key areas of unmet need and ongoing debate:

### 8.1. Lack of Contemporary Multicenter Randomized Controlled Trials (RCTs)

The landmark randomized, sham controlled trial by Weinshenker et al. remains the primary high-quality evidence base for TPE in MS [30]. However, it predates advances in immunopathological understanding, imaging technology, and clinical trial methodology. No large-scale, multicenter RCTs have since been conducted using modern diagnostic criteria, advanced imaging endpoints (e.g., volumetric MRI or central vein sign), or fluid biomarkers. Consequently, most supporting data derive from retrospective and uncontrolled designs, limiting the ability to differentiate true treatment-related improvements from spontaneous relapse remission and hindering generalizability to contemporary clinical practice.

### 8.2. Heterogeneous Definitions of Corticosteroid-Refractory Relapse

Operational definitions of treatment failure after high-dose corticosteroids vary widely, ranging from lack of subjective improvement to persistence of neurological deficits at arbitrary timepoints (e.g., 3–14 days). This inconsistency introduces selection bias, complicates patient stratification, and limits the comparability across studies (Table 4). A unified consensus-based definition of steroid-refractory relapses is essential to harmonize research protocols and clinical practice.

### 8.3. Inconsistent Outcome Measures and Limited Use of Composite Endpoints

Most studies rely on the Expanded Disability Status Scale (EDSS) [63] or visual acuity as primary outcomes. While validated, these measures have limited sensitivity to detect subtle but clinically meaningful changes in motor, cognitive, and sensory domains. Incorporating composite metrics such as the Multiple Sclerosis Functional Composite (MSFC) [64] or patient-reported outcomes (e.g., MSIS-29, Neuro-QoL) [65,66,67] would provide a more nuanced evaluation of treatment efficacy and improve the translation relevance of study findings.

### 8.4. Understudied Role in Progressive MS with Superimposed Relapses

The therapeutic role of TPE has been primarily evaluated in RRMS, whereas patients with progressive phenotypes may also experience acute superimposed relapses. The immunological characteristics and responsiveness of these events are poorly defined. Prospective studies focused on this subpopulation could clarify the utility of TPE in progressive MS and support development of individualized treatment algorithms [68].

## 9. Integration into Clinical Practice

Therapeutic plasma exchange has been endorsed by major neurological and transfusion societies as an effective intervention for managing severe corticosteroid-refractory relapses in MS. The AAN recommend TPE as an adjunctive therapy in treatment-resistant relapses, particularly when neurological deficits are disabling or vision-threatening [28].

The ASFA classifies TPE as a Category I indication, denoting first-line therapy, for acute demyelinating diseases such as MS when there is inadequate response to high-dose intravenous corticosteroids [29]. In clinical practice, a critical prerequisite before initiating TPE is the confirmation that the event represents a genuine MS relapse rather than a pseudo-exacerbation or an alternative demyelinating disorder, as inappropriate escalation may expose patients to procedural risks without therapeutic benefit.

Despite these formal endorsements, the real-world implementation of TPE remains suboptimal, particularly outside of tertiary or academic referral centers. Barriers include limited access to apheresis units, insufficient training among general neurologists, heterogeneity in institutional protocols, and constraints related to reimbursement, cost, and hospital resources. These challenges are particularly pronounced in community hospitals and low-resource settings.

Successful integration of TPE into clinical practice for MS relapses requires four strategic pillars. First, rigorous patient selection, prioritizing cases with severe or disabling relapses, lesion localization in functionally eloquent regions (e.g., optic nerve, spinal cord), evidence of active inflammation (gadolinium-enhancing lesions), documented corticosteroid resistance, and consideration of comorbidities. Second, streamlined workflows must ensure early identification of non-responders and rapid initiation of TPE (ideally within 14 days of relapse onset) to maximize recovery potential. Third, interdisciplinary collaboration among neurologists, transfusion medicine, and experienced nursing staff to ensure procedural safety and optimize efficiency. Finally, expanding targeted educational initiatives for clinicians to reduce knowledge gaps and foster appropriate utilization.

Equally important is a clear operational definition of the severe MS relapses warranting TPE. These typically include optic neuritis with profound visual loss, (≤0.3 Snellen), acute myelitis producing paraplegia/tetraplegia or sphincter dysfunction, disabling brainstem syndromes (e.g., bulbar dysfunction, severe ataxia), or motor relapses leading to loss of independent ambulation. The presence of gadolinium-enhancing lesions on MRI further strengthens the rationale for TPE, particularly when corticosteroid resistance has been documented.

## 10. Future Directions

Despite growing evidence supporting the role of TPE in managing corticosteroid-refractory MS relapses, several pivotal avenues require further exploration to optimize its clinical utility and broaden its impact.

### 10.1. Comparative Efficacy Trials

There is a pressing need for multicenter randomized controlled trials (RCTs) that directly compare TPE with other rescue therapies such as IVIG, immunoadsorption (IA), and emerging targeted immunotherapies, including anti-IL-6, anti-CD19, and FcRn inhibitors [69,70,71]. Such trials should employ standardized definitions of corticosteroid–refractory relapse, harmonized treatment protocols, and validated composite outcome measures that incorporate both functional recovery and imaging biomarkers. Stratification by relapse type (e.g., optic neuritis, myelitis) and disease course (relapsing vs. progressive) will be essential to determine the most effective rescue modality for specific clinical scenarios.

### 10.2. Biomarker Discovery and Validation

Development of predictive biomarkers remains a critical unmet need. Integration of high-throughput proteomic, transcriptomic, and metabolomic analyses with advanced neuroimaging modalities such as quantitative susceptibility mapping, myelin water imaging, PET tracers of inflammation, may help delineate biological subtypes of MS relapses most responsive to apheresis-based interventions. Candidate biomarkers include serum neurofilament light chain (sNfL), complement activation profiles, CSF cytokine signatures, or autoantibody repertoires. Longitudinal biobanking and systems biology approaches will be critical for correlating biomarker dynamics with clinical outcomes, ultimately guiding precision selection for TPE [70,72,73,74].

### 10.3. Economic and Health Systems Research

Robust health-economic evaluations are needed to determine the cost-effectiveness of TPE, both as a stand-alone therapy and in combination regimens, particularly in resource-constrained healthcare systems. Analyses should incorporate not only direct procedural costs, but also long-term disability outcomes, healthcare utilization, and quality-adjusted life years (QALYs). Cost-utility models comparing TPE to second-line immunotherapies or prolonged hospitalization may inform reimbursement frameworks and health policy decisions. In parallel, comparative studies assessing organizational models for delivering TPE, such as centralized vs. decentralized apheresis centers, could guide optimal resource allocation [75].

### 10.4. Combination and Sequential Therapeutic Strategies

Emerging data suggest that combining or sequencing TPE with targeted immunomodulatory agents may enhance therapeutic efficacy. For example, sequential use of TPE followed by anti-CD20 monoclonal antibodies (e.g., rituximab, ocrelizumab, or ofatumumab) could achieve rapid clearance of pathogenic immunoglobulins and sustained suppression of their production. Trials evaluating optimal timing, dosing schedules, and immunological rebound phenomena are warranted [76]. Further research is required to define optimal timing, dosing schedules, and the risk of immunological rebound. Another priority is clarifying the ideal treatment window: while most evidence supports initiation within 7–14 days of relapse onset, experience from NMOSD and pooled MS/NMOSD cohorts suggests that very early initiation may yield superior outcomes. This hypothesis, however, requires validation in MS-only populations.

## 11. Conclusions

Therapeutic plasma exchange has emerged as a critical intervention in the management of MS relapses unresponsive to high-dose corticosteroids. Evidence from observational studies and select randomized controlled trials demonstrates meaningful recovery, particularly in patients with severe attacks, short disease duration, and active MRI lesions. Although its precise mechanisms remain under investigation, TPE is thought to exert benefit through rapid removal of pathogenic humoral factors, including autoantibodies, immune complexes, and complement components, that are not effectively targeted by corticosteroids. Importantly, this effect is transient reflecting modulation of acute antibody mediated inflammation.

Despite its favorable safety profile when administered in experienced centers, TPE remains underutilized globally due to logistical constraints, protocol variability, and limited awareness among providers. While current guidelines from the AAN, and ASFA endorse its use as a first line escalation therapy for steroid-refractory relapses, implementation often lags due to insufficient infrastructure, a lack of standardized referral pathways, and disparities in access, particularly in low- and middle-income settings.

Persistent knowledge gaps include the absence of contemporary multicenter randomized controlled trials incorporating modern imaging and composite functional outcomes, heterogeneous definitions of steroid-refractoriness, and the lack of validated biomarkers to guide patient selection. The therapeutic role of TPE in progressive MS with superimposed relapses also remains insufficiently defined. Future priorities should focus on patient stratification using advanced biomarker platforms, optimization of combination regimens (e.g., TPE followed by B-cell depleting therapies), rigorous cost-effectiveness evaluations, and the development of equitable access programs. Establishing harmonized protocols, interdisciplinary care models, and educational initiatives will be essential to ensuring consistent and timely use of TPE.

In summary, TPE represents an indispensable, though underutilized, therapeutic option in the acute management of steroid-refractory MS relapses. With continued research and concerted efforts to standardize and expand access, TPE has the potential not only to improve short-term recovery but also to meaningfully influence the long-term outcomes in carefully selected patient populations.

## Figures and Tables

**Table 2 biomedicines-13-02399-t002:** Predictors of positive and negative responses to therapeutic plasma exchange.

Category	Predictor	Effect of Response
Demographics	Age < 45 years	Positive
Disease duration	<5 years	Positive
Clinical phenotype	Optic neuritis, myelitis	Positive
MRI findings	Gadolinium enhancement	Positive
Time initiation	<14 days	Positive
Baseline EDSS	>7.5	Poorer outcome
No MRI activity	Absent of Gadolinium enhancement	Limited benefit

MRI: Magnetic resonance imaging; EDSS: Expanded Disability Status Scale.

**Table 3 biomedicines-13-02399-t003:** Common adverse events and management strategies in therapeutic plasma exchange.

Adverse Event	Incidence	Mechanism	Mitigation
Hypotension	9–23%	Volume shift	Slow infusion, fluids
Hypocalcemia/metabolic alkalosis	0.3–7.8%	Citrate anticoagulation	Calcium supplementation
Nausea/Dizziness	11–18%	Volume/electrolyte shifts	Symptomatic treatment
Catheter infections	<10%	Noncompliance with aseptic technique during the insertion and maintenance of intravascular catheters	Aseptic technique, early removal
Allergic reactions	3–12% using FFP 0.2–0.3% using albumin	Recipient’s immune response to foreign proteins in the FFP	Premedication

FFP: fresh frozen plasma.

**Table 4 biomedicines-13-02399-t004:** Multiple sclerosis relapses refractory to high-dose corticosteroids.

Reference	Criterion	Definition/Threshold
Bunganic et al., 2022 [8]	Timeframe	No clinical improvement within 10–14 days after the final dose of IVMP
Trebst et al., 2009 [59]	Neurological status	No meaningful improvement in EDSS or functional domains (motor, sensory, vision)
Kleiter et al., 2018 [33]	Optic Neuritis	Persistent visual acuity ≤ 0.3 (Snellen 20/70 or worse) after 2 weeks post-IVMP
Weinshenker et al., 1999 [30]	Motor/sensory systems	Failure to recover strength or gait function in clinically affected limbs
Ehler et al., 2015 [35]	Physician assessment	Patient remains with moderate/severe disability, defined as no return to baseline function or <1-point EDSS gain

## Data Availability

No new data were created or analyzed in this study.

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
