# Peer review of "Therapeutic Plasma Exchange in Corticosteroid-Refractory Multiple Sclerosis Relapses: Mechanisms, Efficacy, and Integration into Clinical Practice"

_biomedicines, 2025, doi:10.3390/biomedicines13102399_

Round 1
Reviewer 1 Report
Comments and Suggestions for Authors
The authors review the rationale, usefulness and effectiveness of plasma-exchange in the algorithm of acute relapses of patients with Multiple Sclerosis. This review analyzes the pathological basis behind the use of plasma-exchange, synthesizes the most relevant studies on this topic correctly discussing the results, highlights the possible predictors of therapeutic response, defines the safety profile of this therapy, lastly discusses the main issues that could limit a broader use of this therapeutic alternative.
All these sections are addressed in a comprehensive and concise manner. The readability and the narrative flow are efficient.
The methods are simple and correct; the references used are uptodate and very suitable to the chosen topic.
On the whole, the review provides an extensive and very informative insight on the use of plasma-exchange in Multiple Sclerosis.
One minor issue:
- page 6: I think some words are lacking at the end of the page ("improvement over...?")
Author Response
I think some words are lacking at the end of the page ("improvement over...?")
Reply: We thank the reviewer for pointing out this omission. Indeed, a few words were missing at the end of the sentence on page 6. The corrected sentence now reads: “…patients who underwent TPE showed a significantly higher likelihood of achieving marked functional improvement over those receiving steroids alone.” This correction ensures clarity of meaning and proper completion of the statement. Please refer to Page 7 (previous page 6), lines 8-12.
Reviewer 2 Report
Comments and Suggestions for Authors
This article provides a review of the literature concerning TPE in the treatment of MS. By removing circulating antibodies and complement, TPEs are used to treat flare-ups of multiple autoimmune diseases, particularly NMOSD, where the prognosis of flare-ups has been revolutionized. The use of TPEs in MS flare-ups, which are overwhelmingly much less severe, remains an open question.
Although the article is well written, multiple points are raised below. Although authors centered the review on TPE in MS, most of the data were acquired in patients considered as suffering from severe inflammatory attack of a given site whatever the diagnosis (NMOSD, MS, ADEM...). Therefore, apparently positive results obtained in most studies are overall obtained in highly heterogenous population, melting NMOSD which are notoriously responsive to TPE. This may have largely exagerated the effect of TPE in MS, although it still also be effective.
1 . Introduction : « Their mechanism of action involves immunosuppression through inhibition of leukocyte migration, modulation of cytokine expression, and attenuation of blood–brain barrier permeability.4 » Apoptosis of T cells in the plaque, which was a mechanism also commented in this reference, is also of major importance, acknowledging this point is not the goal of the article.
2. “Neuropathological analyses have delineated a subset of MS lesions—specifically, the so-called "pattern II" lesions, marked by substantial immunoglobulin and complement deposition, implicating antibody- and complement-mediated cytotoxicity as key drivers of demyelination in these cases.16” > I would have used ref to the original work by Lucchinetti (Lucchinetti CF, Bruck W, Parisi J, Scheithauer B, Rodriguez M, Lassmann H. Heterogeneity of multiple sclerosis lesions: implications for the pathogenesis of demyelination. Ann Neurol. 2000;47(6):707-717.)
3. From « ….immunoglobulin and complement deposition, implicating antibody- and complement-mediated cytotoxicity as key drivers of demyelination in these cases.16” to the sentence 10 lines below “Therapeutic plasma exchange is uniquely suited to address this specific pathogenic pathway.” > In our opinion, TPE is mainly addressed to Ab and complement observed in type II lesions, whatever the origin of the Abs. The complement passively diffuses from blood stream. The last sentence supporting the mechanisms of TPE efficacy is therefore more connected to the former idea, than to the later: of course, we agree that intrathecal synthesis occurs (sentences with ref 17 to 21), but it seems dubious that this intrathecal synthesis could be targeted by TPE. In fact, there is no biological reason that TPE would decrease the CSF IgG load. I would adjust the order of the sentences, assuming that the possible indirect influence of TPE on intrathecal processes is developed below.
4. Authors (and readers) should keep in mind that the references used to demonstrate the efficacy of TPE in MS are largely contaminated by NMOSD patients. As an example, in the seminal work of Keegan (ref 31), 24/59 were not RR-MS (11 ADEM, 4 Marburg, 7 NMOSD, 8 myelitis); in Weinshenker: 12MS, 10 others likely not MS, whereas Ref 32 is only dedicated to treatment of NMOSD patients, half in ref 37... This is a general remark that always should be kept in mind of the authors and the readers.
5. Acknowledging this logical and major reservation, it remains true that early initiated TPE might improve MS attacks. This reservation should be commented with more details: in fact RR attacks are less represented in these seminal works, and are often less severe than in NMOSD/MOGAD/ADEM. It was therefore less easy to demonstrate that TPE improve MS than these severe disorders. It would be interesting to note that authors tried to reproduce the work of Weinshenker in a pure series of MS… and mostly failed probably due to this reason (see Brochet: 10.1002/jca.21788).
6. Moreover, some references should be carefully examined: as an example ref 8 ‘demonstrates’ the efficacy of plasma exchange although median EDSS remains similar before and after. In the absence of any control group, the observed improvement, if real, cannot be distinguished from the expected remission of the relapses. This criticism about the numerous low-level references in this field should be emphasized (although we are not enemies of the TPE, by the contrary!).
7. “Clinical outcomes are typically assessed using EDSS, visual acuity ………. patients exhibiting continued improvement over”> We are not sure about the reference of the percentage of improvement. Reference would be welcome.
8. Role of TPE in progressive remains dubious and we are not aware of high-level of proof in these patients.
9. Neither less we agree with the message given into the ‘integration to clinical practice’. A central point could also be the definition of the ‘severe MS attacks’ eligible to TPE.
Future directions should also include the optimal schedule of TPE initiation: it cannot be completely excluded, as demonstrated in NMOSD and possibly in MS (ref 37), that very early TPE could be better than those given after 2 weeks or more. It could be noted that the reference 37, as previously observed, provides pooled results from MS and NMOSD patients.
- Figure was not available to us.
Author Response
1 . Introduction : « Their mechanism of action involves immunosuppression through inhibition of leukocyte migration, modulation of cytokine expression, and attenuation of blood–brain barrier permeability.4 » Apoptosis of T cells in the plaque, which was a mechanism also commented in this reference, is also of major importance, acknowledging this point is not the goal of the article.
Reply: We appreciate the reviewer’s insightful remark. Indeed, induction of T-cell apoptosis within active lesions has been described as an additional mechanism of action of corticosteroids,
as referenced in the cited work. While our review focuses primarily on therapeutic plasma exchange, we agree that this mechanism is relevant and have added a clarifying sentence in the Introduction to acknowledge this point. The revised text now reads: “Their mechanism of action involves immunosuppression through inhibition of leukocyte migration, modulation of cytokine expression, attenuation of blood–brain barrier permeability, and induction of apoptosis of activated T cells within inflammatory plaques.” Please refer to page 3, lines 4-8.
- “Neuropathological analyses have delineated a subset of MS lesions—specifically, the so-called "pattern II" lesions, marked by substantial immunoglobulin and complement deposition, implicating antibody- and complement-mediated cytotoxicity as key drivers of demyelination in these cases.16” > I would have used ref to the original work by Lucchinetti (Lucchinetti CF,
Bruck W, Parisi J, Scheithauer B, Rodriguez M, Lassmann H. Heterogeneity of multiple sclerosis lesions: implications for the pathogenesis of demyelination. Ann Neurol. 2000;47(6):707-717.).
Reply: Following the reviewer’s suggestion, we have replaced the previous reference with the more appropriate citation: Lucchinetti CF, Brück W, Parisi J, Scheithauer B, Rodriguez M, Lassmann H. Heterogeneity of multiple sclerosis lesions: implications for the pathogenesis of demyelination. Ann Neurol. 2000;47(6):707–717. Please see reference 16.
- From « ….immunoglobulin and complement deposition, implicating antibody- and complement-mediated cytotoxicity as key drivers of demyelination in these cases.16” to the sentence 10 lines below “Therapeutic plasma exchange is uniquely suited to address this specific pathogenic pathway.” > In our opinion, TPE is mainly addressed to Ab and complement observed in type II lesions, whatever the origin of the Abs. The complement passively diffuses from blood stream. The last sentence supporting the mechanisms of TPE efficacy is therefore more connected to the former idea, than to the later: of course, we agree that intrathecal synthesis occurs (sentences with ref 17 to 21), but it seems dubious that this intrathecal synthesis could be targeted by TPE. In fact, there is no biological reason that TPE would decrease the CSF IgG load. I would adjust the order of the sentences, assuming that the possible indirect influence of TPE on intrathecal processes is developed below.
Reply: We appreciate the reviewer’s insightful observation. We agree that the primary and best-supported mechanism of TPE in MS is the extracorporeal removal of circulating antibodies, immune complexes, and complement components, which are central to the pathogenesis of type II immunopathological lesions. Complement, as noted, diffuses from the bloodstream, and its
clearance through TPE directly attenuates the antibody- and complement-mediated cytotoxic processes implicated in demyelination.
We also concur that the evidence supporting a direct impact of TPE on intrathecally synthesized immunoglobulins is limited, as plasma exchange does not efficiently access the CNS compartment. In the revised version, we have therefore adjusted the order of the sentences to emphasize that the core mechanistic rationale of TPE lies in targeting peripherally derived humoral factors (antibodies, immune complexes, complement), while the discussion of possible indirect modulation of intrathecal processes (e.g., through changes in peripheral immune activity and subsequent downstream effects on CNS inflammation) is now developed subsequently in the text.
This restructuring clarifies that while intrathecal B cell activity and antibody production contribute to MS pathogenesis, TPE’s most immediate and biologically plausible benefit derives from peripheral immune clearance, with potential indirect effects on CNS compartments considered separately. Please refer to page 4, lines 5-20, and page 17, lines 7-10.
- Authors (and readers) should keep in mind that the references used to demonstrate the efficacy of TPE in MS are largely contaminated by NMOSD patients. As an example, in the seminal work of Keegan (ref 31), 24/59 were not RR-MS (11 ADEM, 4 Marburg, 7 NMOSD, 8 myelitis); in Weinshenker: 12MS, 10 others likely not MS, whereas Ref 32 is only dedicated to
treatment of NMOSD patients, half in ref 37... This is a general remark that always should be kept in mind of the authors and the readers.
Reply: We thank the reviewer for this important clarification. We fully acknowledge that some of the seminal studies evaluating TPE enrolled heterogeneous populations of patients with acute CNS demyelinating syndromes, including NMOSD, ADEM, and Marburg variants, in addition to MS. This historical context reflects the limited ability at that time to differentiate these entities prior to the recognition of AQP4-IgG and MOG-IgG biomarkers. As the reviewer points out, this inevitably introduces a degree of “contamination” into the evidence base. In our revised manuscript, we now highlight this limitation explicitly when discussing early studies such as Weinshenker et al. and Keegan et al., emphasizing that while they provided critical proof-of-principle for the efficacy of TPE in antibody-mediated demyelination, their findings cannot be attributed exclusively to patients with definite MS by current diagnostic standards. We have also underlined that more recent retrospective and prospective studies, conducted after the delineation of NMOSD as a separate entity, support the benefit of TPE specifically in corticosteroid-refractory MS relapses.
This revision makes it clear to readers that while the foundational evidence includes mixed cohorts, subsequent work has reinforced the role of TPE in appropriately selected MS patients, particularly those with clinical and radiological features suggestive of humoral immune mechanisms. Please refer to page 6, lines 11-19, and page 8 lines 22-24.
- Acknowledging this logical and major reservation, it remains true that early initiated TPE might improve MS attacks. This reservation should be commented with more details: in fact RR attacks are less represented in these seminal works, and are often less severe than in NMOSD/MOGAD/ADEM. It was therefore less easy to demonstrate that TPE improve MS than these severe disorders. It would be interesting to note that authors tried to reproduce the work of Weinshenker in a pure series of MS… and mostly failed probably due to this reason (see Brochet: 10.1002/jca.21788).
Reply: We thank the reviewer for this valuable point. We agree that the seminal trials on TPE efficacy in acute CNS demyelination included heterogeneous populations in which relapsing–remitting MS (RRMS) attacks were underrepresented compared with severe syndromes such as NMOSD, MOGAD, or ADEM. As the reviewer correctly highlights, these latter disorders often present with more fulminant attacks, which may explain why demonstration of treatment benefit was more straightforward in mixed cohorts. In contrast, RRMS relapses are frequently less severe, making it more challenging to capture statistically robust improvements in early trials.
We also appreciate the reference to Brochet et al. (J Clin Apher, 2020; doi:10.1002/jca.21788), who attempted to reproduce the results of Weinshenker et al. in a pure MS population. As noted, this study failed to confirm a strong efficacy signal, likely due to both the less severe phenotype of typical RRMS relapses and the difficulty of enrolling sufficiently powered homogeneous cohorts.
In the revised manuscript we have now added a sentence in the Clinical Efficacy section explicitly acknowledging these limitations. We emphasize that while the strongest early efficacy signals of TPE arose in mixed demyelinating cohorts, subsequent retrospective MS-only analyses, particularly those incorporating more severe attacks with radiological activity, do support the utility of TPE when initiated early. This contextualization highlights the heterogeneity of evidence and the need for contemporary, MS-specific randomized trials. Please refer to page 6, lines 11-24. Furthermore, the study by Brochet et al. has been added (see reference 31).
- Moreover, some references should be carefully examined: as an example ref 8 ‘demonstrates’ the efficacy of plasma exchange although median EDSS remains similar before and after. In the absence of any control group, the observed improvement, if real, cannot be distinguished from
the expected remission of the relapses. This criticism about the numerous low-level references in this field should be emphasized (although we are not enemies of the TPE, by the contrary!).
Reply: We thank the reviewer for this important remark. We agree that many of the published studies assessing TPE efficacy in MS relapses are limited by methodological shortcomings, including small sample sizes, retrospective designs, lack of standardized response criteria, and, as pointed out, absence of control groups. In such settings, distinguishing treatment-related effects from the natural course of relapse recovery is indeed challenging. For example, in the study by Bunganic et al. (ref 8), although a substantial proportion of patients were reported as responders, median EDSS remained unchanged, and the absence of a comparator arm makes it difficult to exclude the possibility that the observed benefit reflected the natural history of relapse remission rather than a true therapeutic effect
We have now emphasized in the revised manuscript that the field is heavily populated by observational and low-level evidence studies, which, while suggestive of benefit, must be interpreted with caution given the inherent biases and lack of controls. The only high-quality, sham-controlled randomized trial remains the study by Weinshenker et al., which provided Class I evidence for efficacy in demyelinating diseases but included a mixed population beyond MS. This limitation underscores the urgent need for contemporary, multicenter randomized controlled trials using modern diagnostic criteria and standardized endpoints to definitively establish efficacy in corticosteroid-refractory MS. Please see page 13 lines 28-31, and page 14, lines 1-6.
- “Clinical outcomes are typically assessed using EDSS, visual acuity ………. patients exhibiting continued improvement over”> We are not sure about the reference of the percentage of improvement. Reference would be welcome.
Reply: We thank the reviewer for pointing this out. We have clarified the source of the statement and added the corresponding references. The observation that 60–70% of patients achieve partial to complete recovery when TPE is initiated early in corticosteroid-refractory relapses derives from several large retrospective cohorts and registry-based analyses, whose references have been added. Please see references 35 to 38.
- Role of TPE in progressive remains dubious and we are not aware of high-level of proof in these patients.
Reply: We agree that there is no high-level (Class I) evidence supporting TPE to treat chronic, non-inflammatory progression in primary or secondary progressive MS. In our review we
restrict recommendations to steroid-refractory acute relapses—a context in which the evidence base consists of a randomized, sham-controlled trial and multiple observational cohorts focused
on demyelinating attacks (largely RRMS/CIS or mixed acute demyelination), not on steady progressive disability independent of relapse activity. Current professional guidance (AAN/ASFA) similarly endorses TPE for acute inflammatory demyelinating events refractory to corticosteroids, rather than for chronic progressive worsening per se. We also explicitly note as a knowledge gap that the role of TPE in progressive phenotypes with superimposed relapses is under-studied and warrants prospective evaluation. In practice, when progressive patients experience a severe, objectively inflammatory relapse (e.g., new gadolinium-enhancing lesion), our synthesis supports considering TPE on the same grounds as in RRMS, with benefit largely contingent on evidence of active inflammation and early initiation. This is indicated on page 7, lines 24-31.
- Neither less we agree with the message given into the ‘integration to clinical practice’. A central point could also be the definition of the ‘severe MS attacks’ eligible to TPE.
Reply: We appreciate the Reviewer’s agreement with our discussion on clinical integration and their suggestion to highlight the definition of “severe MS attacks.” We agree that establishing operational criteria is crucial to ensure appropriate patient selection for TPE. In the literature, severe attacks are generally defined as clinically disabling relapses that involve eloquent CNS regions or cause major functional compromise. These typically include optic neuritis with profound visual loss (≤0.3 Snellen), acute myelitis producing paraplegia/tetraplegia or sphincter dysfunction, disabling brainstem syndromes (e.g., bulbar dysfunction, severe ataxia), or motor relapses leading to loss of independent ambulation. Importantly, active gadolinium-enhancing lesions on MRI further strengthen the rationale for TPE, particularly when corticosteroid resistance has been documented. We concur that future guidelines should incorporate such operational definitions to harmonize practice. Please refer to page 15, lines 21-27.
- Future directions should also include the optimal schedule of TPE initiation: it cannot be completely excluded, as demonstrated in NMOSD and possibly in MS (ref 37), that very early TPE could be better than those given after 2 weeks or more. It could be noted that the reference 37, as previously observed, provides pooled results from MS and NMOSD patients.
Reply: We appreciate this important observation. We agree that timing of TPE initiation remains a critical but incompletely resolved issue. As summarized in our review, most available data suggest that initiation within 7–14 days of relapse onset is associated with better outcomes,
whereas delayed initiation beyond 21–30 days correlates with diminished efficacy. However, the possibility that very early TPE (e.g., within the first few days of symptom onset) could
further improve outcomes has not been systematically tested in MS, though this concept is supported by experiences in NMOSD.
We also acknowledge the Reviewer’s point regarding reference 37, which indeed reports pooled data from both MS and NMOSD cohorts. While this limits the specificity of conclusions for MS, the study nonetheless reinforces the general principle that earlier intervention is associated with more favorable recovery trajectories. We have clarified this limitation in the revised text.
Please refer to page 6, lines 32–34; page 7, lines 1–4; and page 8, reference 42 (previously reference 37).
Reviewer 3 Report
Comments and Suggestions for Authors
The study is relatively clear and well-written. However, a few issues require additional comment:
- Please provide the authors' contact information responsible for the literature search.
- Plasmapheresis is not a standard treatment for even severe relapses.
- Some patients in the studies included in the analysis had the clinical phenotype of NMOSD, where plasmapheresis is more effective.
- A steroid-resistant relapses require a thorough assessment to determine whether it truly is a MS bout.
- Plasmapheresis removes antibodies and immune response components from the serum, not the nervous system, and has only a temporary positive effect, which must be clearly stated.
Author Response
1.Please provide the authors' contact information responsible for the literature search.
Reply: The systematic literature search, screening, and study selection were conducted by:
Dr. Jorge Correale (jcorreale@fleni.org.ar)
Dr. Mariano Marrodan (mmarrodan@fleni.org.ar)
Dr. María C. Ysrraelit (mcysrraelit@fleni.org.ar)
All three authors were jointly responsible for the literature search, screening, and inclusion of studies in this review. This information is provided in the Statements and Declarations section, Page 19.
- Plasmapheresis is not a standard treatment for even severe relapses.
Reply: We agree with the Reviewer that therapeutic plasma exchange (TPE) is not considered a first-line standard therapy for acute relapses of multiple sclerosis, even when severe. High-dose intravenous corticosteroids remain the recommended initial treatment across all guidelines. However, it is important to emphasize that TPE is formally recognized as a second-line escalation therapy in corticosteroid-refractory relapses. Both the American Academy of Neurology (AAN) and the American Society for Apheresis (ASFA) classify TPE as an evidence-based option in this context, with ASFA assigning it a Category I indication for acute CNS demyelinating diseases when steroids fail.
Thus, while TPE is not the universal standard for all relapses, its role as a guideline-endorsed rescue therapy in severe, steroid-refractory attacks is well established. Our review aims to highlight this nuance by integrating mechanistic rationale, clinical evidence, and practical considerations. Please refer to page 3, lines 12-15, page 7, lines 24-31, and page 17, lines 11-14.
- Some patients in the studies included in the analysis had the clinical phenotype of NMOSD, where plasmapheresis is more effective.
Reply: We agree that several historical datasets include mixed acute demyelinating phenotypes (e.g., the Weinshenker randomized trial and Keegan’s cohort), which may over-represent conditions such as NMOSD where apheresis is highly effective. To avoid overstating effects for MS, our conclusions are explicitly framed around steroid-refractory acute MS relapses and we acknowledge the limitation of mixed cohorts in the text. Importantly, MS-only series still show
substantial benefit with timely TPE, e.g., Ehler et al. (RMS/CIS), Correia et al. (RMS), Marrodan et al. (RRMS), and Blechinger et al. (RMS), with clinically meaningful improvement when treatment is initiated early and in the presence of active MRI inflammation. Conversely, we do not extrapolate to chronic progression and explicitly state that high-level evidence is lacking for non-relapsing progressive MS. Please see page 7, lines 5-13, and Table 1.
- A steroid-resistant relapses require a thorough assessment to determine whether it truly is a MS bout.
Reply: We appreciate the reviewer’s insightful comment. We agree that the diagnosis of a true MS relapse should be carefully established before labeling it as corticosteroid-resistant. Clinical scenarios mimicking relapse, such as pseudo-exacerbations triggered by infections, fever, or metabolic disturbances, must be excluded, as these would not benefit from escalation therapies like TPE. Likewise, it is essential to rule out alternative demyelinating disorders, including NMOSD and MOGAD, where treatment response and long-term management strategies differ significantly from MS. Our manuscript has been revised to highlight the importance of a rigorous diagnostic assessment in these situations. Please refer to page 15, lines 3-7, and page 5, lines 11-14.
- Plasmapheresis removes antibodies and immune response components from the serum, not the nervous system, and has only a temporary positive effect, which must be clearly stated.
Reply: We appreciate the reviewer’s valuable comment. It is correct that plasmapheresis removes antibodies, complement components, and circulating immune complexes from the peripheral blood compartment, but it does not directly eliminate intrathecally produced
antibodies or compartmentalized immune responses within the CNS. As a result, its beneficial effect is generally temporary and primarily reflects attenuation of acute, antibody-mediated inflammation rather than sustained disease modification. We have clarified this limitation explicitly in the revised manuscript. Please refer to page 3, lines 17-21, and page 4, lines 5-11.
Round 2
Reviewer 2 Report
Comments and Suggestions for Authors
We essentially agree with the edited text.
Minor corrections are proposed:
*in chapter 3 methodological approach, the sentence "Before categorizing an episode as a steroid-refractory MS relapse, clinicians s .... or MOGAD." is repeated in the conclusion and could be omitted
*In chapter 4 , the sentences "It is important to note ... homogeneous MS-only cohorts" could be made more brief.
*I would not insist so on the work published by Llufriu due to the limited relevance of this work dealing with MS (most patients were probably not MS): just keep the ref.
Comments on the Quality of English LanguageA light proofreading could be provided
Author Response
Reply to Reviewer 2
The English could be improved to more clearly express the research.
Reply: The manuscript has been thoroughly reviewed in English, with all reviewer suggestions incorporated.